# Unravelling the drivers of marine biodiversity across the Phanerozoic

**Alexis Balembois** [1] ✉, **Alexandre Pohl** [2], **Bertrand Lefebvre** [3], **Thomas Servais** [4], **Daniel J. Lunt** [5], **Paul J. Valdes** [5] & **Grégory Beaugrand** [1] ✉

Understanding the potential drivers of spatial-temporal patterns in biodiversity has been a central tenet in biogeography and palaeontology for decades. More than 30 hypotheses have been proposed, including null models and theories based on environmental controls, energy, area, speciation/extinction dynamics, time, habitat features, ecological niches and biotic interactions. Yet, no consensus has been reached, and question remains whether a primary cause explains temporal trends and spatial patterns in biodiversity such as the latitudinal biodiversity gradient. Here we combine a macroecological model with global climate simulations to show that the niche-environment interaction may explain changes in global marine biodiversity and associated large-scale spatial patterns during the Phanerozoic (last 541 million years). We show that the niche-environment interaction imposed both a species carrying capacity and spatial constraints on marine biodiversity that defined the latitudinal biodiversity gradient. Although our model suggests that climate modulated the niche-environment interaction, hence spatial biodiversity patterns, it also demonstrates that palaeogeographical evolution imposed changes in shallow-water area, continental fragmentation and the location of landmasses relative to climatic belts and may have constituted the fundamental driver of changes in global marine biodiversity at the geological time scale. Therefore, several mechanisms interacted to balance the niche-environment interaction and drove the trajectory of marine biodiversity during the Phanerozoic.

Identifying the mechanisms driving the global pool of species on Earth and its spatial distribution has mobilized the scientific community for decades[1,2]. Among the most studied global patterns in biodiversity, the latitudinal biodiversity gradient (LBG), which consists today in an increase in biodiversity from the poles to low latitudes[3], has been widely investigated in the marine and terrestrial realms and more than 30 hypotheses or theories have been proposed, mainly in the field of biogeography[1,2,4] but also in

palaeontology[5–7]. Yet, although it was considered as one of the 25 most important scientific enigmas in 2005[8], the origin of spatial biodiversity patterns still remains unsolved. Another aspect that remains contentious is whether the LBG originates from a primary cause or results from the interaction of multiple factors[1,5].

Proposed hypotheses or theories to explain large-scale biodiversity patterns such as the LBG have ranged from null/neutral (e.g., mid-domain effect[9] and the neutral theory of biodiversity and

[1]Univ. Littoral Côte d'Opale, CNRS, Univ. Lille, UMR 8187 LOG, Wimereux, France. [2]Université Bourgogne Europe, CNRS, Biogéosciences UMR 6282, Dijon, France. [3]Université Claude Bernard Lyon 1, ENSL, CNRS, LGL-TPE, Villeurbane, France. [4]CNRS, Univ. Lille, UMR 8198 Evo-Eco-Paleo, Lille, France. [5]School of Geographical Sciences, University of Bristol, Bristol, UK. ✉e-mail: alexis.balembois@univ-littoral.fr; gregory.beaugrand@cnrs.fr

biogeography[10]) to phenomenological (e.g., models involving abiotic factors such as temperature and nutrients[11,12]) and mechanistic (e.g., origination/extinction dynamics[13,14] models). Mechanisms have included area[15], energy (e.g., exo- or endo-somatic energy[16-18]), origination/extinction dynamics and evolution, time[19], habitat features[20], niche theories[21] and biotic interaction (e.g., competition-predation trade-off[22]). Among abiotic parameters, temperature has often been proposed to play an important role either directly[11,12] or indirectly through its influence on the structure of the water column[23]. These hypotheses or theories are reviewed in Supplementary Note 1 and Supplementary Fig. 1.

Hypotheses proposed to explain spatial-temporal changes in marine biodiversity have mostly been suggested in isolation based on observations of the modern ocean. The present day provides excellent data sets[24] but a limited range of environmental conditions specific to the ongoing interglacial period. Yet, abundant work suggests that global biodiversity and underlying spatial patterns (e.g., LBGs) have significantly varied over geological time[6,25]. Valid biogeographical explanations should also hold at the geological time scale[26]. The palaeontological record thus constitutes an unparalleled opportunity to test hypotheses established in the modern.

Here, we couple a macroecological model with general circulation model (GCM) simulations to investigate the mechanisms that drove changes in global marine biodiversity (i.e. the global pool of species) and spatial marine biodiversity patterns during the Phanerozoic (last 541 million years). We show that our macroecological model satisfactorily reproduces both spatial biodiversity patterns in the modern ocean and the first-order temporal changes in global marine biodiversity during the Phanerozoic, in agreement with palaeontological databases. Then, we use this simulated spatial-temporal reconstruction of marine biodiversity to investigate the drivers of spatial-temporal trends in marine biodiversity. We analyse the contributions of ocean temperature, landmass spatial distribution, shallow-shelf area, continental fragmentation and allopatric speciation to simulated trends. Our results demonstrate that the niche-environment interaction (NEI), which expression in our model was modulated through time by palaeogeography and global climate, may have imposed the evolution of global marine biodiversity and its spatial patterns during the Phanerozoic.

## Results and discussion
### Simulating marine biodiversity during the Phanerozoic
We base our simulations of marine biodiversity during the Phanerozoic on an updated version of the species niche and climate interaction (SNCI) model[27] (named SNCI-v2 hereafter; Methods). This model has been built as part of the MacroEcological Theory on the Arrangement of Life (METAL)[27-29]. METAL combines principles from biogeography, macroecology, evolutionary biology, and community ecology, offering a unified perspective on the arrangement of biodiversity[27-29]. This theory considers that the niche-environment interaction (NEI) is fundamental to understand the arrangement of biodiversity at different organisational levels. The niche summarises emergent properties at the individual level that originate from the phenotypic expression of the genome (e.g. life history and physiological traits) within a given environment, and the NEI reflects the dynamic interplay between the niche and the environmental regime[30]. At the individual level, the NEI affects individual's behaviour, such as thermotaxis[30]; at a species level, it controls the abundance of a species in space and time (e.g. phenological and biogeographical shifts); at a community level, it affects community organisation and controls biodiversity arrangement[31].

Our global, spatially-resolved macroecological model SNCI, based on the NEI, has been shown to successfully capture spatial marine biodiversity patterns in the modern, recent past and deep geological past (last glacial maximum, 21,000 years ago[28] and Cambrian to Ordovician, > 445 Ma[25]). It permits to simulate biodiversity in the absence of robust constraints on the environmental affinities of extinct marine species that once populated the ocean and provides insights of global biodiversity patterns through the Phanerozoic, such as LBGs. The model generates a pool of 100,000 virtual niches sensu Hutchinson[32], which then interact with the local environmental conditions (here, monthly sea surface temperature (SSTs)) to give potential spatial pseudo-species ranges. Without allopatric speciation, one niche is always associated with one pseudo-species[31]. When allopatric speciation is considered in the model, a niche can generate more than one pseudo-species if spatial pseudo-species ranges are fully separated (Methods). The rationale supporting this implementation of allopatric speciation is that the full spatial separation of pseudo-species ranges interrupts the gene flow between the distant populations that gradually accumulate genetic differences, which, after several million years (the amount of time separating our subsequent time slices being in the order of 5 million years), gives rise to distinct species. Thereafter, our simulations include allopatric speciation, unless otherwise stated.

Before analysing changes in biodiversity simulated in the geological past, we start by assessing the robustness of our simulation for the present day. The highly significant correlation of spatial marine biodiversity (i.e. species richness) patterns simulated for 0 Ma with those from a previously-validated version of the SNCI[27] model, as well as with observations (foraminifera and fish)[24], demonstrates that our macroecological model, forced with SSTs from the HadCM3[33] climate model, captures well present-day large-scale spatial biodiversity patterns, including the modern LBG (Fig. 1). Although simulated pseudo-species represent undefined categories of organisms in line with previous work[27], our model captures well biodiversity patterns of major groups of vertebrates (fish, Fig. 1c) and protists (foraminifera, Fig. 1d). Spatial auto-correlation is unlikely to have affected our results as minimum degrees of freedom for the correlation to remain significant were far lower than actual degrees of freedom ($df^* = 2$, $df^* = 4$ and $df^* = 4$ between SNCI-v2 and SNCI-v1, SNCI-v2 and fish data and SNCI-v2 and foraminifera data, respectively) (Methods).

In line with previous work[34] we focus our investigation of marine biodiversity changes during the Phanerozoic on the shallow-water environments around (palaeo)continents (defined as model grid cells adjacent to landmasses in our main simulations), because they represent the main part of the Phanerozoic palaeontological databases. In our main simulations, environmental constraints (SSTs) driving the macroecological model were provided by the ocean-atmosphere GCM HadCM3[33]. This coupled atmosphere-ocean-vegetation model is widely applied in palaeoclimate reconstructions to simulate past and future changes. A total of 109 climatic simulations were available over the whole Phanerozoic from Valdes and colleagues[35], every ca. 5 million years; the palaeogeographical reconstructions used in these simulations are from the PALEOMAP project[36] and the atmospheric $CO_2$ reconstruction from Foster and colleagues[37] (Methods). The range of the tropical SSTs in these HadCM3 simulations, between 23.7 °C and 32.7 °C (mean of 28.77 °C, standard deviation $\sigma = 1.92$ °C, $n = 109$), is at the lower end of proposed estimates (Supplementary Fig. 2). For each of the 109 climatic simulations available, providing us with SST and bathymetry data, we calculate an equilibrium marine biodiversity using our macroecological model, which leads to a series of 109 simulated reconstructions of marine biodiversity through time (Fig. 2). The same pool of 100,000 virtual niches was used in all simulations.

When our macroecological model is run over the Phanerozoic, simulated changes in marine biodiversity are significantly correlated positively with long-term temporal trends reconstructed based on fossil data ($r = 0.73$, $p < 0.01$, $p_{ACF} = 0.03$, $df = 107$, Fig. 3a, b). To reduce the impact of the biases (e.g., preservation and sampling) present in each individual fossil database, we performed a Principal Component Analysis (PCA) on the fossil-based biodiversity curves of

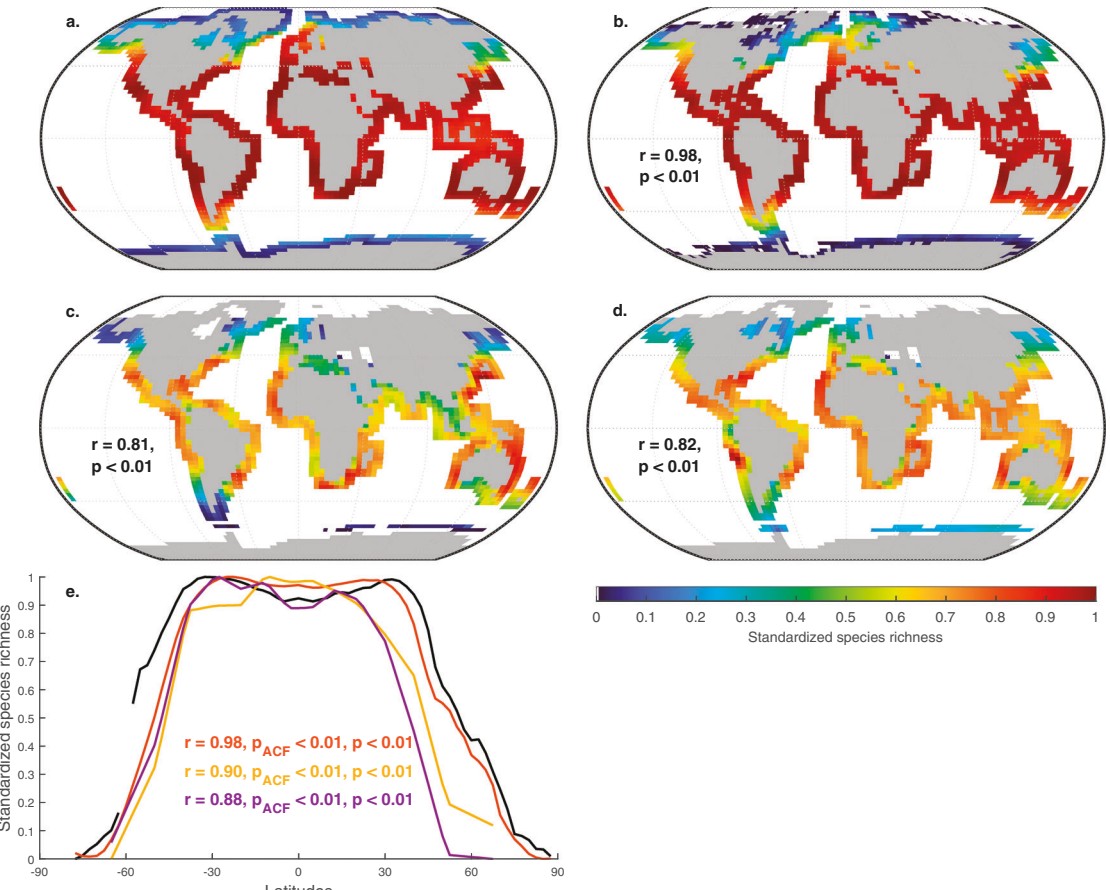

**Fig. 1 | Comparison of simulated modern marine biodiversity spatial patterns with previous models and data. a** Marine biodiversity simulated using our updated macroecological model (SNCI-v2; Methods) forced with a HadCM3 simulation for 0 Ma. Robinson projection with parallels shown every 30° latitude. Emerged landmasses are shaded gray. **b** As per **a** using a previous version of the SNCI model[27]. The coefficient r and its p-value for the correlation with SNCI-v2 results (data shown in **a**) are provided. **c** Map of biodiversity for fish[24] and correlation with SNCI-v2 results (**a**). **d.** Map of biodiversity for foraminifera[24] and correlation with SNCI-v2 results (**a**). **e** Standardised LBG simulated in SNCI-v2 (solid black line; see **a**) and a previous version of the SNCI model[27] (orange line; see **b**) and standardised LBG for fish[24] (purple line, see **c**) and foraminfera[24] (yellow line; see **d**). LBGs are calculated as the mean number of species per latitude. Correlation coefficients r and probabilities for the correlations before (p) and after (p_ACF, ACF for autocorrelation function), accounting for autocorrelation with SNCI-v2 results (solid black line), are provided using the same color code as for lines.

Sepkoski et al.[38], Alroy et al.[39], the PaleoBiology DataBase (PBDB), Zaffos et al.[40], and a recently updated version of the database of Sepkoski et al.[38]. This operation also allows us to combine recent and updated but generalist databases with others that better match our numerical experimentation conditions (Methods). The first component of the PCA captured 81 % of the total variance; it is referred to as fossil species richness index hereafter (Fig. 3a). We find a significant positive correlation between long-term changes in simulated global biodiversity (when including allopatric speciation) and the fossil species richness index (Fig. 3b and Supplementary Table 1). In particular, the model reproduces well the long-term increase in biodiversity during the Ordovician ('Great Ordovician Biodiversification Event'[41]), subsequent minimum during the Triassic and the pronounced increase in fossil biodiversity observed during the Jurassic and the Cretaceous.

Our simulations and the fossil species richness index diverge during the Cenozoic. While modelled global biodiversity flattens off at the end of the Cretaceous, the fossil species richness index keeps increasing. This mismatch may represent limitations in our (climatic and/or macroecological) simulations and/or a decrease in fossil preservation or sampling back in geological time[42,43]. The ability of our model to capture the strong increase in biodiversity between the Jurassic and the Cretaceous, without invoking biological innovation, suggests an environmental driver to the Mesozoic marine revolution[44],

in line with previous work[45]. Our model, however, deviates from Cermeño and colleagues[45] in suggesting that an environmentally-modulated species carrying capacity strongly contributed to shaping the evolution of marine biodiversity during the Phanerozoic, as discussed below. Although mass extinctions play a well-known role in the evolution of life and therefore in the composition of communities[46], the ability of our coupled climate-macroecological model to capture first-order trends in Phanerozoic marine biodiversity suggests that marine extinctions–not explicitly resolved nor externally imposed in our simulations–did not critically alter the long-term trajectory of global marine biodiversity. Our results suggest that, at the multi-million-year timescale, biodiversity is in quasi-equilibrium with environmental conditions. While ecological changes, exemplified by several biotic transitions separating mega-assemblages[47] over the last 541 million years, may reflect the legacy of past extinctions, global biodiversity may not reflect such legacy.

### Analysing the drivers of global biodiversity changes

We use our spatial-temporal simulation of Phanerozoic marine biodiversity, validated against modern spatial patterns (Fig. 1) and long-term temporal trends (Fig. 3b), to analyse the drivers of biodiversity changes over the last 541 million years. We first quantify the contribution of allopatric speciation to long-term biodiversity changes by repeating our main simulations without allopatric speciation (Fig. 3c).

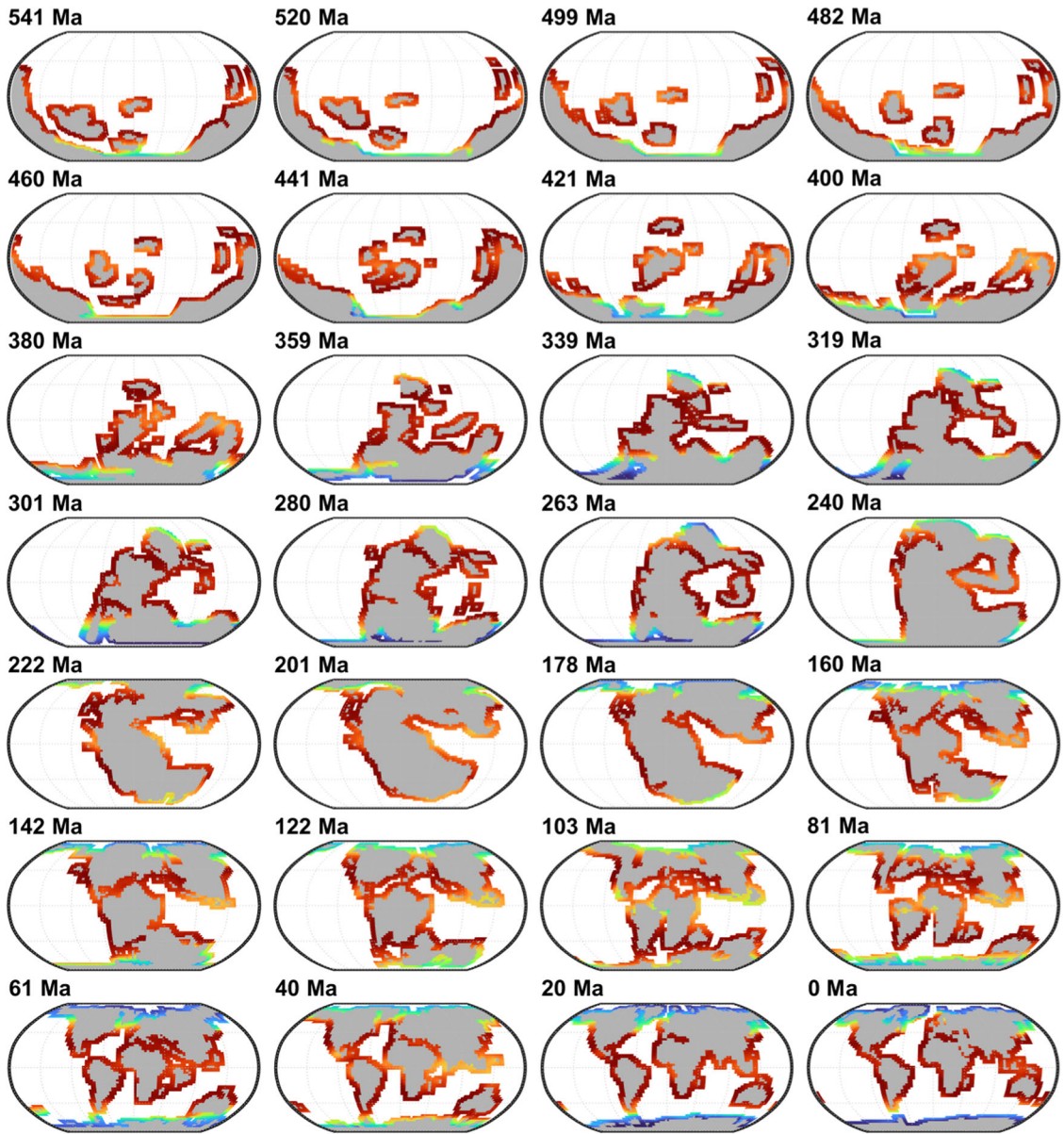

**Fig. 2 | Global maps of simulated marine biodiversity during the Phanerozoic.** Only 28 out of 109 time slices[35] are shown here every ca. 20 Ma, while all time slices are used in our analyses. Robinson projection with parallels shown every 30° latitude. Emerged landmasses are shaded gray. See Supplementary Movie 1 for a more complete view of species richness maps through the Phanerozoic.

Analyses excluding allopatric speciation lead to identical large-scale biodiversity patterns (Figs. 1 and 2); allopatric speciation only impacts global biodiversity estimates (Methods). Modelled changes in global biodiversity without allopatric speciation are not statistically correlated with the fossil species richness index ($r = 0.08$, $p = 0.40$, $p_{ACF} = 0.80$, degree of freedom df = 107, Fig. 3c), demonstrating that allopatric speciation played a key role in the evolution of global marine biodiversity during the Phanerozoic (Fig. 3b).

Next, we seek to understand what environmental parameters drove (allopatric speciation hence) temporal trends in biodiversity during the Phanerozoic. We find a significant positive correlation of the fossil species richness index with marine area around continents ($r = 0.75$, $p < 0.01$, $p_{ACF} = 0.03$, df = 107, Fig. 3d) and to a lesser extent with the continental fragmentation index[40] ($r = 0.57$, $p < 0.01$, $p_{ACF} = 0.14$, df = 107, Fig. 3e). We note that the latter correlation is not significant at the threshold of 0.05 after accounting for temporal autocorrelation, suggesting that long-term dynamics drive the correlation rather than the higher-frequency variability. Area is an important

component of the equilibrium theory of insular biogeography[48] and habitat fragmentation has often been proposed among the key drivers of species richness (see Supplementary Fig. 1 and Supplementary Note 1). Area and the index of continental fragmentation are highly positively correlated for the Phanerozoic (Supplementary Table 1). Area enhances spatial environmental heterogeneity, which, together with habitat fragmentation, increases the probability for allopatric speciation because greater environmental heterogeneity increases environmental islands[49].

We then examine whether the joint influence of the location of the continental masses and the LBG induced by the NEI influenced global marine biodiversity. To do so, we calculated two indices (Methods). Referred to as the latitudinal continental index (LCI) hereon, the first index is merely the sum of the number of marine cells adjacent to continental masses with at least one species for each latitude and for all months (Methods). The second index is the LCI weighted by the LBG (i.e. number of cells weighted by the average latitudinal value of modelled species richness). Strong relationships are observed

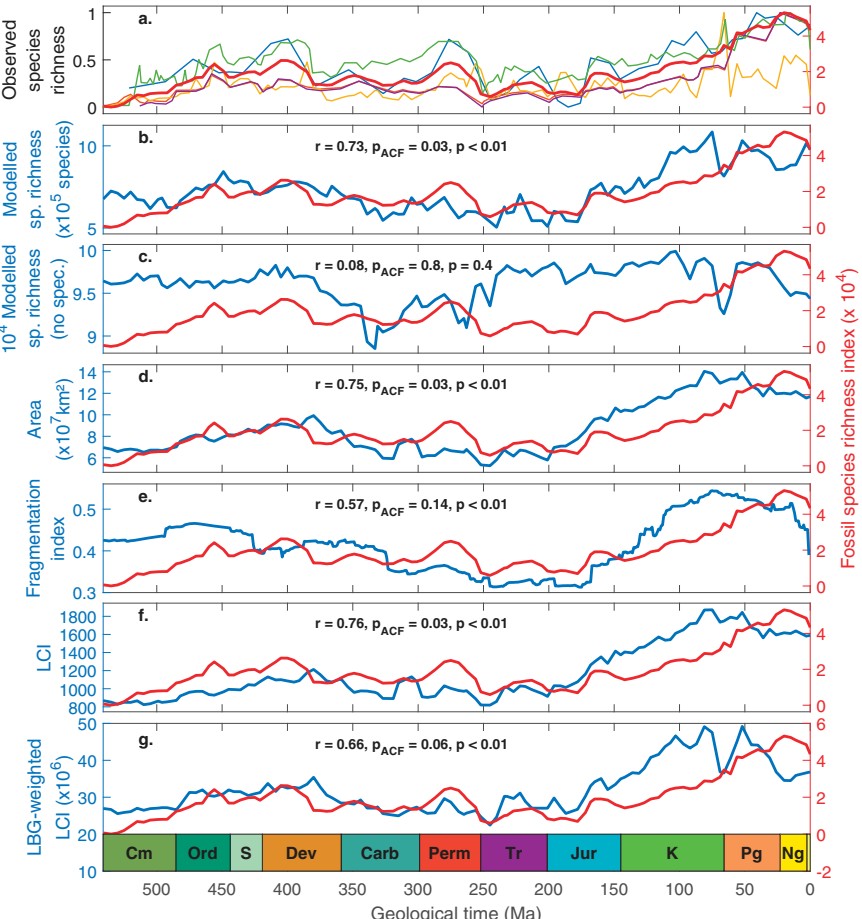

**Fig. 3 | Temporal trends in marine biodiversity during the Phanerozoic. a** Long-term changes in observed species richness; blue, orange, yellow, purple and red thin lines are from Alroy[39], a recently updated version of the Sepkoski's database, PBDB, Sepkoski[38], and Zaffos[40], respectively. The resulting first principal component, referred to as fossil species richness index (see main text and Methods), is the thick, red line. **b** Long-term changes in HadCM3-based simulated biodiversity (blue) in relation with the fossil species richness index; allopatric speciation was included in the model. **c** As per panel b without allopatric speciation. **d** Long-term changes in available marine shallow-water area in relation with the fossil species richness index. **e** Long-term changes in the continental fragmentation index (blue) in relation with the fossil species richness index. The continental fragmentation index is after ref. [40]. **f** Long-term changes in the latitudinal continental index (LCI, blue) in relation with the fossil species richness. **g** Long-term changes in LBG-weighted LCI (blue) in relation with the fossil species richness index. The correlation coefficient r, and its probability before (p) and after ($p_{ACF}$, ACF for autocorrelation function) accounting for temporal autocorrelation are provided in each panel. Cm Cambrian, Ord Ordovician, S Silurian, Dev Devonian, Carb Carboniferous, Perm Permian, Tr Triassic, Jur Jurassic, K Cretaceous, Pg Paleogene, Ng Neogene. Ma million years ago. Sp. Richness species richness. No spec. no speciation. Fossil spe richness ind fossil species richness index.

between the two LCI indices and the fossil species richness index until the end of the Cretaceous (Fig. 3f, g). From the end of the Cretaceous onwards, the trends of the fossil species richness index and both LCIs start to differ. Long-term changes in the LBG-weighted LCI are also strongly correlated with long-term changes in modelled global biodiversity, which suggests that this process strongly drove global species richness in our macroecological model ($r = 0.9$, $p < 0.01$, $p_{ACF} < 0.01$, Supplementary Fig. 3 and Supplementary Table 1). This analysis suggests that, for a large part of the Phanerozoic, changes in global biodiversity were driven by the interplay between the location of the main continental masses and the NEI that affects the position of the LBG.

To confirm this observation, we compare LBG shapes with the location of main continental masses and global temperatures through time (Fig. 4). Clear relationships appear between LBGs (Fig. 4a) and continental drift (Fig. 4b), as landmasses were only located in the Southern Hemisphere at the beginning of the Phanerozoic and progressively moved northwards. In a climatic scenario with smooth temperature variations through time, such as the one considered in our main simulations based on temperature fields of Valdes and colleagues[35] (Fig. 4c and Supplementary Fig. 2), LBGs steepen when most landmasses are located in the low latitudes and become

smoother when landmasses are more uniformly distributed in space. Additionally, LBGs are truncated in the Northern Hemisphere until *circa* 360 Ma when continental masses start to be widespread in this hemisphere (Fig. 4b). The steepness of the LBG strengthens during cold climatic regimes (Fig. 4c), showing an equatorial (or low-latitude) maximum at around 345 Ma, 260 Ma and 250 Ma (the temporal resolution used in our modelling does not permit capturing rapid environmental perturbations such as global warming during the Permian-Triassic extinction). In contrast, the LBG flattens during warmer periods, being often associated with an equatorial reduction in species richness. This analysis corroborates the importance of the interaction between climate, the location of the main continental masses and the NEI in controlling marine biodiversity spatial arrangement during the Phanerozoic.

**Sensitivity testing**
We have shown that large-scale spatial biodiversity patterns strongly depend on the location of the main landmasses and their interaction with regional climate. We here test the sensitivity of our results to the climate model, SST scenario and palaeogeographical reconstruction. We use two additional series of HadCM3 simulations, and three

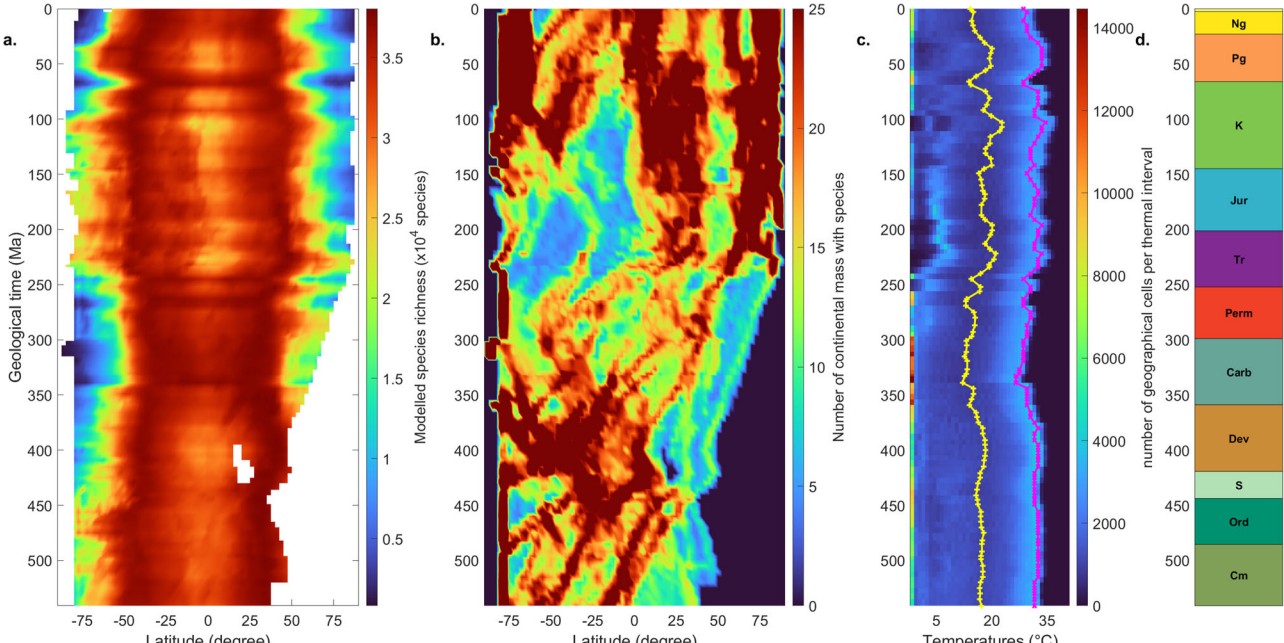

**Fig. 4 | Long-term changes in LBG simulated using the HadCM3 SSTs of ref. 35.**
**a** Long-term changes in latitudinal species richness. White latitudinal cells correspond to latitudes and times with no continent. **b** Number of continental mass cells in which at least one species can establish in adjacent marine cells per latitude through time. **c** Long-term changes in the density distribution (number of cells) along temperature categories from −2 °C to 41 °C (water temperatures) for intervals of 1 °C. Fuchsia and yellow curves are long-term changes in the temperature interval where the maximum of observations is found and the mean of temperatures, respectively. **d** Geological chart. Cm Cambrian, Ord Ordovician, S Silurian, Dev Devonian, Carb Carboniferous, Perm Permian, Tr Triassic, Jur Jurassic, K Cretaceous, Pg Paleogene, Ng Neogene. Ma million years ago.

additional series of simulations using the Earth system model of intermediate complexity cGENIE[50]. cGENIE is a biogeochemistry-enabled ocean general circulation model[51] coupled with an energy-moisture-balance atmospheric model and a sea-ice model. Our cGENIE simulations use a lower spatial resolution (36 × 36 equal-area grid) and only 28 time slices through the Phanerozoic (Methods), as shown for example in Supplementary Figs. 4, 5, 6 and 7.

The first series of additional HadCM3 simulations is similar to the series of Valdes and colleagues[35], but includes several changes to the underlying climate model, notably tuned atmospheric and ocean physics to give latitudinal temperature gradients in better agreement with palaeotemperature proxies[52], and to the imposed Cenozoic $CO_2$ concentrations. Indeed, while the original series from Valdes and colleagues[35] used Cenozoic $CO_2$ concentrations after Foster and colleagues[37], Cenozoic $CO_2$ concentrations in this new series of simulations follow Rae and co-workers[53]. The second series[52] of additional simulations is similar to the previous one except that it uses changes in $CO_2$ concentration through the Phanerozoic that are tuned to give global surface temperatures in line with reconstruction of Scotese and colleagues[54]. As simulations based on these alternative versions of HadCM3 led to similar results and conclusions as our main simulations, we did not include them in this paper.

The first series of cGENIE simulations reproduces the long-term SST trend of the HadCM3 simulations of Valdes and colleagues[35] (see Supplementary Fig. 2) and uses the same continental reconstructions[36], hence showing the impact of the climate model. The second series of cGENIE simulations uses another climate scenario designed to reproduce the long-term SST trend of Grossman and Joachimski[55], still using the same continental reconstructions[36], constituting a sensitivity test to the climate scenario. Indeed, in the first set of palaeoclimatic reconstructions, radiative forcing was adjusted to reproduce the mean tropical (−30 to 30 °N) SSTs of Valdes et al.[35], resulting in smooth mean global temperatures variations ($\sigma = 1.92$) through the Phanerozoic. In the second set of palaeoclimatic reconstructions, radiative forcing was adjusted to fit the tropical SSTs estimated from oxygen isotopes by Grossman and Joachimski[55], leading to stronger global mean SST variations through deep time ($\sigma = 7.11$). In this second reconstruction, temperatures are close to 50 °C during the middle Cambrian, then cool to ~35 °C until the mid-Ordovician (between 500 Ma and 460 Ma), and reach a plateau with moderate temperature variations ($\sigma = 2.20$) between 460 Ma and 360 Ma. A new drop in temperatures occurs between 360 Ma and 340 Ma ($\sigma = 5.48$), and then global mean temperatures become smoother until present days ($\sigma = 3.25$) (see Supplementary Fig. 2). The third series reproduces the long-term SST trend of the HadCM3 simulations of ref. 35 but uses an alternative set of palaeo-geographical reconstructions[45] based on a different geodynamical model (see Supplementary Fig. 7), hence exhibiting differences in the latitudinal position of the landmasses and permitting to quantify the impact of the palaeogeographical reconstruction (Methods).

We first compare long-term trends in modelled species richness based on each of those simulations (Supplementary Fig. 4). Using a model of coarser resolution (cGENIE) while keeping the same climatic scenario and continental configurations (Supplementary Fig. 4b) does not critically affect our biodiversity reconstructions and still gives significant positive correlations with observations, similarly to our HadCM3-based main simulations (Supplementary Fig. 4a). Trends in modelled global biodiversity using similar continental reconstructions with a different climate scenario also show significant correlations with observations (Supplementary Fig. 4c), suggesting that global climate variations (within the range permitted by proxy data) have a limited impact on simulated long-term biodiversity trends. However, global biodiversity simulated based on climatic simulations reproducing the same palaeoclimate trends as Grossman and Joachimski[55] but using alternative continental reconstructions, i.e based on the third series of cGENIE simulations, is not correlated with our fossil species richness index (Supplementary Fig. 4d). Thus, long-term trends in global biodiversity are strongly affected by the location of the continents but are

robust to uncertainties in SSTs. Our results are also little dependent on the numerical model used.

We then analyse long-term changes in LBGs (Supplementary Fig. 5). cGENIE-based simulations reproducing the long-term SST trend of the HadCM3 simulations and using the same continental reconstructions of Scotese and Wright[36] show the same long-term LBGs features as our main, HadCM3-based simulations (Supplementary Fig. 5a vs Fig. 4a), which means that the model spatial resolution does not critically alter spatial biodiversity patterns. Using a climate scenario characterized by higher SSTs during the early Phanerozoic, a drop at the Devonian-Carboniferous boundary (Supplementary Fig. 2), and then a stabilisation at lower values than in the original climate reconstruction, we simulate a quasi-inverted LBG from the Cambrian to ca. 360 Ma followed by a sharper LBG until the end of the Phanerozoic (Supplementary Fig. 5b). The inverted early Phanerozoic LBG is due to the high SSTs that are deleterious to low-latitude marine biodiversity[25]. Conversely, cGENIE-based simulations using a similar climate scenario to the HadCM3 simulations of ref. 35, but another continental reconstruction shows similar long-term LBGs (Supplementary Fig. 5c), suggesting a weak dependence of spatial biodiversity patterns to the position of the continents. Our results, therefore, suggest that while global temperatures have a strong impact on spatial biodiversity patterns, they barely affect long-term changes in global biodiversity. Conversely, landmass location is fundamental to reconstruct long-term changes in the global pool of species.

We also consider the spatial differences with similar climates to show how spatial resolution and continental configurations can affect our spatial biodiversity results (Fig. 2 vs Supplementary Fig. 6 and Supplementary Fig. 7). In Supplementary Fig. 6, we considered similar continental configurations and climates to those in Fig. 2, albeit reconstructed using the cGENIE model providing coarser spatial and temporal resolutions (Methods). We found similar global biodiversity patterns frame by frame; thus, model resolution barely affects spatial global biodiversity patterns. Conversely, in Supplementary Fig. 7, we considered an alternative set of palaeogeographic reconstructions (Methods). While Supplementary Fig. 7 presents different continents location than Fig. 2 and Supplementary Fig. 6, their global biodiversity patterns frame by frame remain similar, giving more strength and confidence to our results. We finally check if different characterisations of marine areas around continents could alter our assessments of long-term changes in global biodiversity (Supplementary Fig. 8). Eight different characterisations have been tested (Methods), and simulated trends are all significantly positively correlated with our fossil species richness index ($0.68 < r < 0.75$, $p_{ACF} < 0.05$). Pair-by-pair correlations (a total of $8 \times 7/2 = 28$ correlations) among long-term changes in global biodiversity based on the eight different characterisations of the marine areas around continents ranged from $r = 0.89$ to $r = 0.99$ ($p_{ACF} < 0.05$), reinforcing our conclusions.

**Putative factors and mechanisms at play**

Our results suggest that the NEI (with niches being based only on SST here) is fundamental in the establishment of a global biodiversity and its large-scale spatial arrangement (Supplementary Note 1). In our model, the NEI generates a mathematical constraint on the maximum pool of niches, and therefore species, which can colonise the inshore ocean. This constraint strongly suggests the existence of an environmental modulation of species carrying capacity both regionally and globally[49,56]. Our results show that the NEI is modulated itself by (i) the latitudinal SST gradient that affects large-scale spatial biodiversity patterns and (ii) the global climatic regime (i.e. greenhouse versus icehouse regime, which may have strongly changed through time)[25] that alters the shape and steepness of a LBG; these two inter-related climatic influences explain why a LBG occurs most of the time during the Phanerozoic but also why it may be altered and even be inverted at some time.

Although other publications have previously shown that a geographical mid-domain effect (i.e., the formation of a biodiversity peak at the middle of a latitudinal range) is unlikely to drive the LBG[27], it does have an effect in the Euclidean space of the niche[27], which means that for the same area, more species are likely to coexist at intermediate (i.e. here around 26 °C) than at much lower or higher temperatures in our macroecological model. Our results provide evidence that the global pool of species is influenced by the NEI in a different way. The location of landmasses (Fig. 3f, g; Fig. 4b and Supplementary Fig. 4) influences the global pool of species. For instance, if all continents were situated in high, cold latitudes, the NEI would result in lower global biodiversity, assuming all other factors and mechanisms remain constant. In addition, available marine area (Category 4 in Supplementary Note 1) around continents enhances allopatric speciation (i) because area increases spatial environmental heterogeneity[49], which creates environmental islands that promote allopatric speciation and (ii) because available marine area around continents is positively correlated with continental fragmentation that favours allopatric speciation through vicariance. Other studies have also suggested eustasy to play an important role in driving Phanerozoic biodiversity as continental shelf flooding generates more ecological niches, thus resulting in more species[57], but fluctuations in sea level are already included in our palaeogeographical reconstructions and we did not investigate this relation further.

We are aware that origination/extinction, historical/time hypotheses and species interaction are important in terms of generation of new species and species saturation[29]. Despite the non-inclusion of these mechanisms in our model, however, our simulations capture both large-scale biodiversity patterns and global marine biodiversity changes. This result suggests that those mechanisms are not leading global biodiversity and its spatial arrangement during the Phanerozoic. Some categories of hypotheses not discussed here (Supplementary Note 1) might also be involved, albeit in a more auxiliary role.

While the present framework cannot quantify the respective influence of these factors, the role that area and continental fragmentation play on the global biodiversity may be substantial[58,59].

**Limitations of our framework**

Although we considered bathymetry and distance to coast, our modelling approach was mainly based upon monthly temperatures. We are aware that the ecological niche sensu Hutchinson is fully multidimensional[60]. Other relevant environmental dimensions include oxygen concentration, food type or quantity, nutrients concentration, solar radiation or photoperiod and the structure of the water column[61]. Temperature is a fundamental driver of biological and ecological systems from the individual to the community organisational level[17]; however, and previous studies have provided evidence that the consideration of temperature alone reproduces well global patterns observed in the fossil records[25,45,62]. In addition, our macroecological model does not include any processes on trophodynamics nor food web and biotic interaction, information that still remains fragmented for most of the Phanerozoic[63]. Such information is important, however, because it may have influenced the emergence and the diversification of some taxonomic groups[64]. We also acknowledge that there are some uncertainties on continental locations[65] that may have influenced our estimate of global biodiversity. Uncertainties may also originate from palaeoclimatic models, palaeoclimatic scenarios and depth of integration of sea temperature (see sensitivity testing). Although this may indeed influence our perception of large-scale spatial biodiversity patterns (Fig. 4), we have shown that this is unlikely to affect substantially our estimate of changes in global biodiversity (Supplementary Fig. 4). Last but not least, our modelled global biodiversity reflects the long-term changes in the whole community. However, differences in long-term changes in species-level biodiversity among taxonomic groups (e.g. Brachiopods, Bryozoan, Sponges, Stromatoporoids and

Corals) have been well documented for some time periods (e.g. from Cambrian to Silurian)[66,67]. Future versions of the macroecological model might provide specific biodiversity patterns for each of these groups.

### Inference from our study

Our macroecological simulations suggest that the NEI controlled the spatial-temporal changes in marine biodiversity during the Phanerozoic. The NEI may have imposed spatial constraints on large-scale spatial biodiversity patterns and determined a global species carrying capacity. Specifically, LBGs emerged from the latitudinal SST gradient, with LBG variations reflecting changes in global climatic state (greenhouse versus icehouse), while simulated global biodiversity changes largely arose from the palaeogeographical evolution. Our simulations suggest that the NEI was hence modulated by climate, the joint influence of landmass location and regional temperature, but also available area, as well as environmental heterogeneity that together promoted allopatric speciation. Therefore, many interacting factors balanced the NEI, indicating that biodiversity results from a multitude of closely imbricated environmental constraints.

## Methods

### Palaeoclimatic simulations

In our main simulations, we use monthly SSTs simulated by Valdes and colleagues[35] using the general circulation model (GCM) HadCM3. Specifically, we used their Phanerozoic series of 109 stage-level simulations based on atmospheric $CO_2$ estimates of Foster et al.[37]. These simulations are based on the coupled atmosphere-ocean-vegetation model HadCM3BL-M2.1aD, which is a variant of the HadCM3 family of models created by the UK Hadley Centre/Meteorological Office[33]. HadCM3 is widely used in global palaeoclimate reconstructions to identify past and future dynamics[33]. The specific version of HadCM3 used in this paper combines an atmosphere model, HadAM3, which has a cartesian grid with a horizontal resolution of $96 \times 73$ grid points (3.75° longitude × 2.5° latitude) and 19 hybrid levels in the vertical, with an ocean component, which has the same horizontal resolution and 20 depth levels which thickness increases from 10 m at the ocean surface to ~600 m at the ocean bottom. HadCM3 also includes a sea-ice model, the MOSES2.1 land-surface exchange scheme, and the TRIFFID vegetation scheme (see the work of Valdes and colleagues[33] for details). In the simulations we used, solar luminosity was varied following the stellar physics of Gough[68], time-specific atmospheric $CO_2$ concentrations were imposed after the reconstruction of Foster and co-workers[37] for 420–0 Ma, extended back to 540 Ma by Valdes and colleagues[35]. The palaeogeography was imposed after reconstructions of the PALEOMAP project[36,65]. For each time period, simulations were run for >5000 model years to ensure that the model was close to thermal equilibrium in the deep-ocean[35]. The simulations are named "Scotese_02" in the nomenclature of Valdes and colleagues[35], and an overview of the simulations is available from https://www.paleo.bristol.ac.uk/ummodel/scripts/html_bridge/scotese_02.html. We led other numerical experiments with "Scotese_07" and "Scotese_08" of Judd and colleagues[52] as well to make sure that the hypotheses used in "scotese_02" did not alter our results and conclusions (unshown). Series 'Scotese_07' is similar to "Scotese_02", but with several modifications to the climate model permitting to simulate latitudinal temperature gradients in better agreement with palaeo temperature proxy data[52]. The $CO_2$ concentrations used as boundary conditions for Cenozoic time slices, after Foster and colleagues[37] in "Scotese_02", were further replaced with those after Rae and colleagues[53]. Series "Scotese_08"[7] is similar to "Scotese_07" except that is uses a $CO_2$ evolution through the Phanerozoic that is tuned to give global surface temperatures following the reconstructions of Scotese and colleagues[54].

We also conducted sensitivity tests to the climatic scenario and palaeogeographical reconstructions, based on new simulations conducted using the carbon-centric Grid Enabled Integrated Earth system model (cGENIE)[50]. cGENIE is an earth-system model of intermediate complexity consisting in a biogeochemistry-enabled ocean general circulation model[51] coupled with an energy-moisture-balance atmospheric model and a sea-ice model. The model was run on a $36 \times 36$ equal-area grid, with 16 depth levels in the ocean. The model turn-around time makes it well adapted for conducting many sensitivity tests, while the model has been shown to satisfactorily simulate ocean biogeochemistry (and circulation) during many geological intervals of the Phanerozoic[69-71]. Our first series of cGENIE simulations was designed to mimic the HadCM3 simulations of Valdes and colleagues[35], by using the same palaeogeographical reconstructions[36] and adjusting radiative forcing in the model to produce identical mean tropical (−30 to 30 °N) SSTs (Supplementary Fig. 2). We conducted one simulation every 20 Ma between 540 Ma and 0 Ma, for a total of 28 time slices, using the cGENIE configurations of Pohl and colleagues[34] (Supplementary Fig. 6). A second series of cGENIE simulations was conducted, featuring an alternative climatic scenario. Radiative forcing was adjusted to reproduce the tropical SSTs reconstructed based on oxygen isotopes by Grossman and Joachimski[55] (Supplementary Fig. 2). The same palaeogeographical reconstructions[36] (hence cGENIE model configurations[34]) were used as in our first series of cGENIE simulations (Supplementary Fig. 6). These simulations are not available for time slices older than 500 Ma because the SST reconstruction does not extend further back in geological time[55]. Our third (and last) series of cGENIE simulations is identical to the first series (reproducing the HadCM3 long-term tropical SST trend[35]) except that it uses alternative palaeogeographical reconstructions (Supplementary Fig. 7). This alternative set of palaeogeographical maps is based on the geodynamical model (hence position of landmasses and deep-ocean bathymetry) of Merdith and co-workers[72] but uses shallow-shelf bathymetry, coastlines and land topography of the PALEOMAP project[36]. Hence, the key difference between the palaeogeographical reconstructions used in our main simulations and these alternative reconstructions is the position of the landmasses, both sets of reconstructions providing very similar estimates in recent geological times but increasingly diverging as we go further back in time (Supplementary Figs. 6 and 7). These alternative reconstructions are described in detail in Cermeño and colleagues[45], of which we used the 30 cGENIE configurations. All cGENIE simulations were run for 10,000 years to reach ocean thermal equilibrium; they were restarted for one additional year with monthly output, and the monthly SSTs were used to force the macroecological model.

### Macroecological modelling

We forced an updated version of the SNCI[27] ("Species-Niche and Climate Interaction") macroecological model with monthly SST data (see previous Methods section) to simulate spatial patterns in biodiversity and long-term changes in global marine biodiversity (i.e. total number of marine species), a key biodiversity metrics[73].

The SNCI model[27] is based on the Macroecological Theory on the Arrangement of Life (METAL)[21], a theory that considers that the niche-environment interaction is fundamental and permits to explain and unify many different patterns and processes observed in biogeography and climate change biology, ranging from the individual to the species and community levels (Supplementary Note 1). The ecological niche *sensu* Hutchinson[32] is the set of environmental conditions that enable a species to grow and reproduce. The SNCI model generates a pool of thermal niches, which then interact with the local temperature regime and allow one (i.e. in the absence of allopatric speciation) or more (i.e. when allopatric speciation is considered) species to colonise the ocean so long as monthly SSTs are suitable. In this work, monthly SSTs were considered for reconstructing biodiversity. Below, we provide a brief

description on how the model works. The version of the SNCI[27] model updated for this study will be referred to as SNCI-v2 hereafter. While all possible ecological niches are generated in SNCI-v1, the second version (SNCI-v2) directly considers a finite pool of randomly-generated niches, reducing the processing time. The reconstruction of large-scale spatial biodiversity patterns and global species richness is carried out in four steps (Supplementary Fig. 9). In Step 1, environmental matrices (i.e. monthly SSTs, continents position, bathymetry) are loaded for all geological periods. We considered marine areas around continents in different ways: (i) one and (ii) two grid cells around continents, one grid cell around continents and bathymetry (iii) <600 m below sea level (b.s.l.) (i.e. threshold), (vi) <1000 m b.s.l., and (v) <1500 m b.s.l. as well as 2 grid cells using the same bathymetry thresholds: (vi) <600 m b.s.l., (vii) <1000 m b.s.l., and (viii) <1500 m b.s.l. These different ways of identifying marine areas around continents did not alter our conclusions (Supplementary Fig. 8). In our main simulations, we used marine areas defined as 2 grid cells around continents.

In Step 2, the model generates the niches. To do so, five parameters $\alpha$, $\beta$, $\gamma$, $\delta$ and $N$ are initialised. They respectively correspond to lower ($\alpha$) and upper ($\beta$) temperature limits of organisms, lower ($\gamma$) and upper ($\delta$) degrees of eurythermy (i.e. minimum and maximum range of thermal tolerance), and the number of independent niches generated ($N$). In the simulation produced for this paper, $\alpha = -2\,°C$, the temperature at which seawater is always frozen, and $\beta = 44\,°C$, the maximum temperature under which a metazoan can reproduce[27]; $\gamma$ was fixed to $4\,°C$ and $\delta$ to $20\,°C$; and $N = 100,000$ (independent) ecological niches. The values of the four parameters are identical ($\alpha$ and $\beta$) or close ($\gamma$) to those used in refs. 25,27,62. ref. 27 performed a sensitivity analysis, and the values are within the range that did not alter large-scale contemporary spatial patterns in biodiversity (their supplementary Table 1). Parameter $\delta$ was halved compared to previous studies[25,27,62] but this is unlikely to strongly affect large-scale spatial patterns in biodiversity because most niches are between $\gamma$ and 46/2, which is about the value of $\delta = 20\,°C$ we chose in this study.

The niches were thereby rectangular-shaped and each was defined as the species thermal range between $(Y - X/2)$ and $(Y + X/2)$, where $X = d1\ x\ (\delta - \gamma) + \delta$ and $Y = d2\ x\ (\beta - X/2 - (\alpha + X/2)) + (\alpha + X/2)$ with d1 and d2 two random scalar values between 0 and 1 introduced to implement variability. Thus, $Y$ corresponds to the centre of the niche and $X$ is the extent of species tolerance range.

In Step 3, ecological niches are projected into a geographical space based on monthly SST data. A total of N presence/absence maps are built for each month per time period. The maps are subsequently summed to get 12 monthly species richness maps per time period. Finally, a mean is applied geographical cell per geographical cell to get a final species richness map per time period.

In Step 4, the 12 monthly maps of presence/absence generated are summed for each niche. When the number of suitable months is greater than or equal to $\varepsilon$ in a geographical cell, the species is considered to maintain annually in that cell; $\varepsilon$ was fixed to 8 months. At the end of the process, a spatial distribution is inferred. When speciation is not considered, one niche gives one species in any case. When allopatric speciation is considered, a discontinuous spatial distribution leads to more than one species, as the number of isolated patches gives the number of potential species resulting from one single niche. While the consideration, or not, of allopatric speciation does not affect large-scale spatial biodiversity patterns, it changes the global number of species for any given period of time. Step 4 leads to the simulated long-term changes in global species richness with or without allopatric speciation.

## Fossil databases

Amongst databases used in this paper, the oldest one is the Sepkoski's database[38]. It focuses on marine biodiversity, especially ichnospecies in marine facies, invertebrate species, marine genera and subgenera and marine metazoan families, through the last 500 Myrs and thus mixes several taxonomic levels. Alroy's database[39] focuses on marine invertebrates genera across the Phanerozoic and is sometimes seen as an update of Sepkoski's database, as it contains more fossils and more sampling conditions. It is also a precursor of the Paleobiodiversity database (PBDB). The updated version of Sepkoski's database has been provided by the *École Normale Supérieure de Lyon* and may be found directly on their website (https://acces.ens-lyon.fr/acces/thematiques/limites/paleobiodiversite/developper/banque-de-donnees-sepkoski). It considers the same marine taxonomic levels as Sepkoski's original database. The PBDB gathers palaeontological data from multiple sources and considers marine species only all across the world for all geological periods. It has been directly downloaded from the PBDB website (https://paleobiodb.org/classic/displayDownloadGenerator) and adapted to fit this study's timeline. Zaffos' database[40] also emanates from the PBDB but only considers marine animal genus richness, which makes comparison with our simulations more reliable. It was originally created to allow long-term comparison with continental fragmentation through the Phanerozoic. While the PBDB is the most recent and updated database here, it is also the most generalist one, and it does not necessarily best fit our numerical experimentation conditions. Moreover, it also includes some biases, for example in preservation and sampling. This is why we chose to combine those databases and to recalculate a fossil species richness index through the Phanerozoic.

## Calculated indices

**Fragmentation index.** The fragmentation index used in this paper is after Zaffos and colleagues[40]. It is defined as the proportion of continental masses connected with other emerged land. A fragmentation index of 0 corresponds to a situation in which all continents are gathered together into one single continental mass (e.g., Pangea). On the contrary, a fragmentation index of 1 means that every plate is separated from the others (i.e. no connection between continents). To build this index, the authors created two maps using the PostGIS (postgis.net) extension of PostgreSQL, one with well separated and ordinary continents configuration and one in which all continents were assembled together. The ratio of the continents' perimeter in the second configuration by the continents perimeter in first map was defined as the fragmentation index. We interpolated the fragmentation index to the same temporal resolution as other data.

**Area around continents.** To calculate the available area (km$^2$) through time, we first identified the grid cells corresponding to this area. Then we used a WGS-84 projection to sort out the area of each geographical cell, and we summed them to get a global available area around continents value per geological period.

**Latitudinal continental index (LCI).** A latitudinal continental index (LCI) was built to evaluate more precisely the influence of continental locations on global species richness. To build this index, for each latitudinal band, we counted the number of marine cells that contained at least one species. The higher the number of continents for a given latitude, the greater the value for that latitude (we remind that the number of marine grid cells around continents was fixed to 2 in our main simulations). Estimates for all latitudes were then added to give the LCI. This index was calculated for each time period.

**LBG-weighted continental index (LBG-weighted LCI).** A second index was also built to consider the joint influence of landmasses and the LBG. To do so, we multiplied the number of marine cells around continents with at least one species for each latitude by the latitudinal value of species richness. The LGB-weighted LCI resulted from the sum

of all latitudinal values. Here, as well, the LBG-weighted LCI was calculated for each time period.

**Long-term changes in observed and modelled species richness**
We used fossil records from 5 databases named accordingly to their respective publication first author: Alroy[39], a recently-updated version of the database of Sepkoski et al.[38], the Paleobiology Database (PBDB; https://paleobiodb.org), Sepkoski[38], and Zaffos[40]. We subsequently standardised them as follows: $X_1 = (X_0 - M_1)/(M_2 - M_1)$, where $X_0$, $X_1$, $M_1$ and $M_2$ were the original value in the data series, the transformed value after 0–1 standardisation, and the minimum and maximum of the original data series, respectively. The goal of the standardisation was to avoid scale effects. We then interpolated the time series at the time scale of our simulations (109 (28–30) time slices for simulations based on HadCM3 (cGENIE)) by means of a Piecewise cubic hermite interpolating polynomial[74,75] method (i.e., a shape-preserving piecewise cubic interpolation method) and applied a standardised principal component analysis[76]. We used the first principal component as a fossil species richness index. The index was then compared to our simulations of global species richness.

**Linear correlations**
Linear correlations were calculated between long-term changes in observed and modelled global species richness and explanatory variables. P-values were calculated both with ($p_{ACF}$) and without ($p$) consideration of autocorrelation between data series. When taking autocorrelation into account, the probability of the correlation coefficient was corrected by adjusting the degree of freedom accordingly to the formula used by Pyper and Peterman[77].

To take into account the potential influence of spatial auto-correlation, we assessed the minimum degree of freedom (df*) needed to keep the correlation significant (i.e. $p \leq 0.05$); a small df* suggests that spatial autocorrelation is less likely to affect the probability of significance[27]. For example, the correlation between biodiversity patterns from SNCI-v2 and the previous version of the SNCI model remained significant for a df* = 2, be a reduction in the degree of freedom of 1343-2 = 1341.

**Reporting summary**
Further information on research design is available in the Nature Portfolio Reporting Summary linked to this article.

## Data availability
All data and code required to repeat the experiments, conduct the analysis of the results and generate the figures are available online at the following link: https://figshare.com/articles/dataset/Code_and_data_repository_for_b_Niche-environment_interaction_controlled_marine_biodiversity_and_its_spatial_arrangement_during_the_Phanerozoic_b_Balembois_et_al_2024_/27201498 and assigned a https://doi.org/10.6084/m9.figshare.27201498 This repository includes the original Matlab codes allowing to run the macro-ecological model as well as fossil species richness curves adapted from Alroy[39], a recently updated version of the Sepkoski's database, PBDB, Sepkoski[38], and Zaffos[40], and palaeoclimatic reconstructions based on the work of Valdes and Colleagues[35] using the continental reconstructions of Scotese and Wright[36], or alternative sets of palaeoclimatic reconstructions using either temperatures based on the work of Grossman and Joachimski[55] or palaeogeographical reconstructions of Cermeño and colleagues[45] (Methods).

## Code availability
All data and code required to repeat the experiments, conduct the analysis of the results and generate the figures are available online at the following link: https://figshare.com/articles/dataset/Code_and_data_repository_for_b_Niche-environment_interaction_controlled_marine_biodiversity_and_its_spatial_arrangement_during_the_Phanerozoic_b_Balembois_et_al_2024_/27201498 and assigned a https://doi.org/10.6084/m9.figshare.27201498 This repository includes the original Matlab codes allowing to run the macro-ecological model as well as fossil species richness curves adapted from Alroy[39], a recently updated version of the Sepkoski's database, PBDB, Sepkoski[38], and Zaffos[40], and palaeoclimatic reconstructions based on the work of Valdes and Colleagues[35] using the continental reconstructions of Scotese and Wright[36], or alternative sets of palaeoclimatic reconstructions using either temperatures based on the work of Grossman and Joachimski[55] or palaeogeographical reconstructions of Cermeño and colleagues[45] (Methods).

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

## Acknowledgements

This is a contribution to UNESCO project IGCP 735 "Rocks and the Rise of Ordovician Life" (Rocks n' ROL). The authors acknowledge the support of the French *Agence Nationale de la Recherche* (ANR) under reference ANR-22-CE01-0003 (project ECO-BOOST) and the support of the Natural Environment Research Council (NERC) through grant NE/X000222/1 (DJL and PJV). We also thank the CPER programme IDEAL (GB). This work was also supported by the graduate school IFSEA that benefits from a France 2030 grant (ANR-21-EXES-0011) operated by the French National Research Agency, and by the *Université du Littoral Côte d'Opale* (ULCO) and the *Institut des Sciences de la Mer et du Littoral* (ISML). The climate model simulations were carried out using the computational facilities of the Advanced Computing Research Centre, University of Bristol - http://www.bristol.ac.uk/acrc/. Calculations were partly performed using HPC resources from DNUM CCUB (*Centre de Calcul de l'Université de Bourgogne*).

## Author contributions

G.B. proposed and designed the study. A.B. and G.B. developed the macroecological model. P.J.V. and D.J.L. carried out the HadCM3 simulations. A.P. performed the cGENIE model simulations. A.B., G.B., A.P., B.L., T.S., P.J.V. and D.J.L. wrote the manuscript with inputs from all of the authors.

## Competing interests

The authors declare no competing interests.
