## [Transparent Peer Review file · Nature Communications]

Unravelling the drivers of marine biodiversity across the Phanerozoic

Corresponding Author: Mr Alexis Balembois

Version 0:

Reviewer comments:

Reviewer #1

(Remarks to the Author)

This manuscript aims to identify the factors driving global spatio-temporal biodiversity patterns through the Phanerozoic. They found that niche-environment interaction explains changes in global marine biodiversity and associated large-scale spatial patterns, specifically by imposing species carrying capacity and spatial constraints. The methodology is one of the most interesting aspects of this study, as it introduces a novel approach to spatio-temporal biodiversity patterns. The approach is interesting and well tested, considering different environmental and geographical configurations.

Below I provide general comments followed by specific line-by-line comments.

The manuscript was hard to read: Overall, the main text needs to be streamlined better. At times, it reads as methods paper, but other times understanding the methodology used was hard from just reading the main text and required switching between the main text and the methodology, disrupting the reading flow. Similarly, the constant comparisons with other papers could be done more aligned with the narrative because as posed, it was rather abstract (the reader should not be expected to go to the references list, find the paper, read it and then make sense of the contrast the authors are referring to). Moreover, the term “niche-environment interaction”, central in this paper, needs to be better defined at the beginning of the manuscript to aid the interpretation of the study. Finally, the Simulating marine biodiversity during the Phanerozoic section could benefit from a schematic figure describing the workflow.

L106-L110 – This paragraph should be better incorporated. In the previous paragraph it was shown that the model performs well using the simulations done by the authors and this paragraph again repeats that it performs well based on other studies. Maybe it can be somehow incorporated into the previous paragraph to create a single paragraph which shows the performance of the model. Alternatively it should be clarified that the test done by the authors focuses on the new version of the model introduced in this study, therefore a new test of performance was needed, if the previous studies talked about in this paragraph used the older version of the model. This should be stated explicitly as it is not clear from the current version of the text.

L116 – not clear what the simulations from ref 34 are, please clarify.

L111-L124 – In this paragraphs there are different names used for time intervals which can cause a bit of a confusion when reading, e.g. “109 simulations”, “109 biodiversity snapshots”, “109 climate simulations”, “all time slices”... It would be good to state that you have looked at 109 time intervals/time slices which were defined by the available 109 climate simulations, or something similar.

L160-L161 – Might be better to say “Analyses excluding allopatric speciation lead to identical large-scale biodiversity patterns as when it is accounted for.” Or similar.

L170-L172 – I am guessing “higher-frequently” should be “higher-frequency”, although I do not fully understand the second half of the sentence.

L176 – “highly correlated positively” should be changed to “highly positively correlated” or “strongly positively correlated”

L181 – LBG being driven by NEI is talked about here, but apart from recovering the modern day LBG in the model testing section, these terms have not been mentioned in the previous sections (apart from the introduction), it feels like it comes out of nowhere.

L202 – “Besides” may not be the best word here, maybe better to use “Additionally”

L208 – “equatorial” has been used in the text previously so “(or low-latitude)” should be incorporated into the first instance of using this word.

L222-L223 – closing bracket is missing

L221-L225 – It is said that simulations were done based on ref 34, then one factor that was adjusted is given in brackets which makes it sound like an example (tuned atmospheric and ocean physics), and then another adjusted factor is given in a separate sentence (imposed Cenozoic CO2 concentrations). This should be streamlined. The wording is also not clear, for example “The imposed Cenozoic CO2 concentrations, originally after ref.36, are also replaced with those of ref.49.” What does the “originally” refer to? Were they based on ref 36 in ref 34? If so, maybe it’s not necessary to mention ref 36? Overall, this paragraph should be a little more streamlined

L232 – I don’t think “additional” is needed here

L232-L233 – maybe it would be good to explain how the two SST trends differ between the two cGENIE simulations (i.e. the SST trends from ref 34 and 51). I see that the differences are explained in the Methods, but for a better clarify of the text a simple explanation should be included in here as well.

L243 – comma before “...main simulations” not needed

L270-L274 – Results presented in Fig. 2, ED Fig. 6 and ED Fig. 7 are being compared and it is highlighted that Results in Fig. 7 show different continent locations than the other two, however it is not clear how Fig. 6 is different and therefore why it is being compared with Fig. 2

L275 – “We finally check that considering different ways to characterise marine areas...” would maybe read better as “We finally check if different characterisations of marine areas...”

L277 – similarly to above instead of “8 ways” maybe it would be better to say “8 characterisations”

L296 – It might be good to explain the mid-domain effect at least briefly

L296-299 – This paragraph reads as an introduction to the following paragraph, therefore should be incorporated accordingly

L313-L317 – This is a very long sentence. I am not sure what “the latter” in L317 refers to.

L320 – “present” is used twice

L343 – I am not quite sure why other studies focusing on biodiversity changes in specific groups are mentioned in this section. Are the findings of these studies significantly different from the patterns uncovered in this study?

Figure 1.e – the text does not look aligned, maybe it has been aligned by the “centre” but it might look better if it’s just aligned to the left

Figure 3. - As the second y-axis shows the same type of value in every subplot just a single “Fossil species richness index” label is needed

Figure 4. legend – in c) it is not totally clear what density distribution refers to (geographical cells?)

Methods

L88-L99 – it is not completely clear how the updated version of SNCI differs from the previous one

(Remarks on code availability)

Reviewer #2

(Remarks to the Author)

Balembois et al. Undertake a project to look at what controls the Latitudinal Biodiversity Gradient (LBG) over the Phanerozoic and long-term changes, using a simulated approach that is then ground-truthed using fossil data. Research about the LBG over the Phanerozoic has gained a lot of interest over the last few years with increasing studies, which have mostly focused on fossil data rather than also simulating hypothetical changes. Overall, I must say I thought the article was excellently written and very easy to follow, and the last section “Limitations of our framework” also captures my main comments. I guess the question is are the limitations too big or are they reasonable. I would view them as future research directions that cannot be fixed quickly, and therefore should not stop this article from being published. I have no major criticisms of the manuscript. I have included below some thoughts I noted as I went through the article, but they are not suggestions for making corrections, more here are some comments and decide if you need to make changes according to them. I should also add that it appears that this model is founded under the idea that a model based on allopatric speciation is acceptable for testing the hypothesis, and I am not qualified to judge this decision.

Comments:

Abstract: It is really strange to see so many references in an abstract. I think they should all be removed and then the ideas should be repeated in the introduction.

Line 26: “whether a single mechanism explains temporal trends and...”. Would you ever expect or believe a study that suggests one mechanism can explain controls on biodiversity? Even in this study it uses an approach that is a model of multiple (not a single) mechanism.

Line 70-71: You look at changes in diversity over the Phanerozoic. Have you ever considered starting in the Ediacaran, especially with the new view that the Cambrian explosion is a series of events that started before the Phanerozoic...

Line 104: Foraminifera are not invertebrates, they are protists. Also seemed odd to not make a comparison to molluscs, bryozoans, echinoderms and brachiopods that have such a rich fossil record.

Line 114: You could have used Triton for investigating the Cenozoic trends in microfossil and the LBG beyond the continental margins.

Line 123: 109 "snapshots" for 541 million years is still quite coarse. I think a higher resolution should be possible.

Line 126: "good job" is a subjective interpretation and funny to see in a scientific article. Could you include a statistical test?

Line 129-130. Only one of the databases in this list is actually up-to-date (PBDB) all of the others are out-of-date and have associated issues that are well-described in the literature. Why not just do the recent download of the PBDB.

Line 140-141: This statement seems odd for the Cretaceous, because compared to older rocks, it is quite well-sampled.

Line 147-150: Do you really believe that mass extinctions, which are partially defined as altering the trajectory in the evolution of life, do not play a first-order role in global biodiversity? I find this statement hard to digest. The modern oceans are more diverse because the Paleozoic fauna were eliminated at the Permian/Triassic boundary and some Mesozoic groups at the Cretaceous/Paleogene boundary. See also what you wrote in Line 205-207.

Line 176-179: Given the experiment set up, this result is a function of the model, right?

William Foster

(Remarks on code availability)

Version 1:

Reviewer comments:

Reviewer #1

(Remarks to the Author)

The authors have addressed all my concerns appropriately and I have no further suggestions. Congratulations!

(Remarks on code availability)

Reviewer #2

(Remarks to the Author)

I have gone through the revised submission and only have a couple tiny correction to highlight, I will also take the opportunity to respond to two rebuttal comments I was not satisfied with, although I do not request further edits as it won't change the article:

line 26: is "drive" the right verb? It sounds funny.

line 31: you are missing a comma before the word hence, otherwise you cannot read the sentence.

line 132: you need a space in-between climaticsimulations.

In the rebuttal you said that you will still use all the fossil databases because they have different biases and by including them you can address this somehow. I find that untrue, first you have the Sepkoski database (and its update), then the Alroy one, and then the PBDB, these are updates on one another fixing issues in the previous whilst providing more data. They all share exactly the same biases, but including them all you are essentially looking at how different amounts of data affect your results, but you could do this by subsampling the PBDB. What Jack Sepkoski and John Alroy achieved is truly phenomenal, but the PBDB is the database to use. P.S., the PBDB is a mix of everything ichnospecies and pollen included.

The second comment I was surprised by was saying that the computer simulations require a lot of time and therefore you cannot provide a higher resolution. The authors come from top universities with great computing facilities and expertise. There is of course a way, through more efficient programming or technological upgrades, that more simulations could be run. I also believe that you are genuinely making the simulations more efficient to run (behind the scenes) as any goal of a computer programmer.

Anyway, it is a good article that deserves to be published.

(Remarks on code availability)

The authors have done a good job of including the code and providing a good README file. I did not install and test the code.

Unravelling the drivers of marine biodiversity across the Phanerozoic

Alexis Balembois^{1*}, Alexandre Pohl², Bertrand Lefebvre³, Thomas Servais⁴, Daniel J. Lunt⁵, Paul J. Valdes⁵, Grégory Beaugrand^{1*}

¹ Univ. Littoral Côte d'Opale, CNRS, Univ. Lille, UMR 8187 LOG, F-62930 Wimereux, France

² Biogéosciences UMR 6282, Université Bourgogne Europe, CNRS F-21000 Dijon, France

³ Université Claude Bernard Lyon 1, ENSL, CNRS, LGL-TPE, F-69622, Villeurbanne, France

⁴ Univ. Lille, CNRS, ULR 8198-Evo-Eco-Paleo, F-59000 Lille, France

⁵ School of Geographical Sciences, University of Bristol, Bristol BS8 1SS, UK

* Corresponding authors: alexis.balembois@univ-littoral.fr, gregory.beaugrand@cnrs.fr

Point-by-point response to Reviewers' comments:

Reviewer 1

Reviewer 1 wrote: *“This manuscript aims to identify the factors driving global spatio-temporal biodiversity patterns through the Phanerozoic. They found that niche-environment interaction explains changes in global marine biodiversity and associated large-scale spatial patterns, specifically by imposing species carrying capacity and spatial constraints. The methodology is one of the most interesting aspects of this study, as it introduces a novel approach to spatio-temporal biodiversity patterns. The approach is interesting and well tested, considering different environmental and geographical configurations.”*

We thank Reviewer 1 for her/his support and for comments that definitively helped to improve the quality of the manuscript.

Reviewer 1 wrote: *“The manuscript was hard to read: Overall, the main text needs to be streamlined better. At times, it reads as methods paper, but other times understanding the methodology used was hard from just reading the main text and required switching between the main text and the methodology, disrupting the reading flow. Similarly, the constant comparisons with other papers could be done more aligned with the narrative because as posed, it was rather abstract (the reader should not be expected to go to the references list, find the paper, read it and then make sense of the contrast the authors are referring to).”*

We understand Reviewer 1's concern. To enhance readability, we have clarified the manuscript in multiple sections to minimize the need for readers to switch between the main text and the methodology, as we agree this may disrupt the reading flow. Additionally, we have added a sentence where necessary to provide context for the references cited. Modifications of the text have been made in lines 86 to 97, lines 98 to 104, lines 137 to 141, lines 254 to 257, lines 262 to 268, lines 272 to 287, lines 326 to 331, lines 360 to 373 and lines 421 to 423 (see the revised manuscript with 'track changes').

Reviewer 1 wrote: “Moreover, the term “niche-environment interaction”, central in this paper, needs to be better defined at the beginning of the manuscript to aid the interpretation of the study. Finally, the Simulating marine biodiversity during the Phanerozoic section could benefit from a schematic figure describing the workflow.”

We have added the following text in the revised manuscript in lines 89 to 97: “*This theory considers that the Niche-Environment Interaction (NEI) is fundamental to understand the arrangement of biodiversity at different organisational levels. The niche summarises emergent properties at the individual level that originate from the phenotypic expression of the genome (e.g. life history and physiological traits) within a given environment and the NEI reflects the dynamic interplay between the niche and the environmental regime¹. At the individual level, the NEI affects individual's behaviour such as thermotaxis¹; at a species level, it controls the abundance of a species in space and time (e.g. phenological and biogeographical shifts); at a community level, it affects community organisation and controls biodiversity arrangement².*”

Reviewer 1 wrote: “L106-L110 – This paragraph should be better incorporated. In the previous paragraph it was shown that the model performs well using the simulations done by the authors and this paragraph again repeats that it performs well based on other studies. Maybe it can be somehow incorporated into the previous paragraph to create a single paragraph which shows the performance of the model. Alternatively it should be clarified that the test done by the authors focuses on the new version of the model introduced in this study, therefore a new test of performance was needed, if the previous studies talked about in this paragraph used the older version of the model. This should be stated explicitly as it is not clear from the current version of the text.”

The location of the paragraph has been modified in the revision. It is now located in lines 98 to 104.

Reviewer 1 wrote: “L116 – not clear what the simulations from ref 34 are, please clarify.”

We have modified the sentence as follows: “*For each of the 109 climatic simulations available, providing us with SST and bathymetry data, we calculate an equilibrium marine biodiversity using our macroecological model, which leads to a series of 109 simulated reconstructions of marine biodiversity through time (Fig. 2).*” in lines 148 to 152. See also lines 140 to 143 where we better described HadCM3-based reconstructions.

Reviewer 1 wrote: “L111-L124 – In this paragraphs there are different names used for time intervals which can cause a bit of a confusion when reading, e.g. “109 simulations”, “109 biodiversity snapshots”, “109 climate simulations”, “all time slices”... It would be good to state that you have looked at 109 time intervals/time slices which were defined by the available 109 climate simulations, or something similar.”

We added “providing us with SST and bathymetry data” in line 149; we changed “The resulting series of 109 ‘biodiversity snapshots’ constitutes our simulated reconstruction of marine biodiversity through time” by “which leads to a series of 109 simulated reconstructions of marine biodiversity through time” in lines 150 and 151; and we changed “time slices” by “simulations” in line 153 to make it clearer for readers.

Reviewer 1 wrote: “L160-L161 – Might be better to say “Analyses excluding allopatric speciation lead to identical large-scale biodiversity patterns as when it is accounted for.” Or similar.“

Changes directly made in the revised version of the text in lines 195 to 197.

Reviewer 1 wrote: “L170-L172 – I am guessing “higher-frequently” should be “higher-frequency”, although I do not fully understand the second half of the sentence.”

We rephrased the sentence as follows (lines 208-211): “...suggesting that long-term dynamics drive the correlation rather than the higher-frequency variability”.

Reviewer 1 wrote: “L176 – “highly correlated positively” should be changed to “highly positively correlated” or “strongly positively correlated”

This has been modified in the revised version of the manuscript in line 214.

Reviewer 1 wrote: “L181 – LBG being driven by NEI is talked about here, but apart from recovering the modern day LBG in the model testing section, these terms have not been mentioned in the previous sections (apart from the introduction), it feels like it comes out of nowhere.”

We added “... and provides insights of global biodiversity patterns through the Phanerozoic, such as LBGs” in lines 102 and 103.

Reviewer 1 wrote: “L202 – “Besides” may not be the best word here, maybe better to use “Additionally”

We changed “besides” by “Additionally” in line 241.

Reviewer 1 wrote: “L208 – *“equatorial” has been used in the text previously so “(or low-latitude)” should be incorporated into the first instance of using this word.*”

We deleted “(or low-latitude)” in line 247 and moved this expression line 243 in the revised text.

Reviewer 1 wrote: “L222-L223 – *closing bracket is missing*”

We deleted the opening bracket line 263 and changed the structure of the end of the sentence, see citation included in our response to the next Reviewer’s comment.

Reviewer 1 wrote: “L221-L225 – *It is said that simulations were done based on ref 34, then one factor that was adjusted is given in brackets which makes it sound like an example (tuned atmospheric and ocean physics), and then another adjusted factor is given in a separate sentence (imposed Cenozoic CO₂ concentrations). This should be streamlined. The wording is also not clear, for example “The imposed Cenozoic CO₂ concentrations, originally after ref.36, are also replaced with those of ref.49.” What does the “originally” refer to? Were they based on ref 36 in ref 34? If so, maybe it’s not necessary to mention ref 36? Overall, this paragraph should be a little more streamlined*”

We rewrote this paragraph in lines 262 to 271 of the revised manuscript: “*The first series of additional HadCM3 simulations is similar to the series of Valdes and colleagues³, but includes several changes to the underlying climate model, notably tuned atmospheric and ocean physics to give latitudinal temperature gradients in better agreement with palaeotemperature proxies⁴, and to the imposed Cenozoic CO₂ concentrations. Indeed, while the original series from Valdes and colleagues³ used imposed Cenozoic CO₂ concentrations after Foster and colleagues⁵, Cenozoic CO₂ concentrations in this new series of simulations follow Rae and co-workers⁶. The second series⁴ is similar to the previous one except that it uses changes in CO₂ concentration through the Phanerozoic that are tuned to give global surface temperatures in line with reconstruction of Scotese and colleagues^{5,4}.*”

Reviewer 1 wrote: “L232 – *I don’t think “additional” is needed here*”

We have deleted the word in the revised text in line 277.

Reviewer 1 wrote: “L232-L233 – *maybe it would be good to explain how the two SST trends differ between the two cGENIE simulations (i.e. the SST trends from ref 34 and 51). I see that the differences are explained in the Methods, but for a better clarify of the text a simple explanation should be included in here as well.*”

We added in lines 279 to 290:

“Indeed, in the first set of palaeoclimatic reconstructions, radiative forcing was adjusted to reproduce Valdes and colleagues³ mean tropical (–30 to 30°N) SSTs, resulting in quite smooth mean global temperatures variations ($\sigma = 1.92$) through the Phanerozoic. In the second set of

palaeoclimatic reconstructions, radiative forcing was adjusted to fit the tropical SSTs estimated from oxygen isotopes by Grossman and Joachimski⁸, leading to stronger global mean SST variations through deep time ($\sigma = 7.11$). In this second reconstruction, temperatures are close to 50°C during the middle Cambrian, then cool to ~ 35°C until the mid-Ordovician (between 500Ma and 460Ma), and reach an equilibrium with moderate temperature variations ($\sigma = 2.20$) between 460Ma and 360Ma. A new drop in temperatures occurs between 360Ma and 340Ma ($\sigma = 5.48$), and then global mean temperatures become smoother until present days ($\sigma = 3.25$) (see Extended Data Fig. 2).”.

Reviewer 1 wrote: “L243 – comma before “...main simulations” not needed”

We have deleted it in the revised version of the manuscript.

Reviewer 1 wrote: “L270-L274 – Results presented in Fig. 2, ED Fig. 6 and ED Fig. 7 are being compared and it is highlighted that Results in Fig. 7 show different continent locations than the other two, however it is not clear how Fig. 6 is different and therefore why it is being compared with Fig. 2”

We added “In Extended Data Fig. 6 we considered similar continental configurations and climates to those in Fig. 2, albeit reconstructed using the cGENIE model providing coarser spatial and temporal resolutions (Methods). We found similar global biodiversity patterns frame by frame; thus, model resolution barely affects spatial global biodiversity patterns. Conversely, in Extended Data Fig. 7 we considered an alternative set of palaeogeographic reconstructions (Methods).” in lines 329 to 334.

Reviewer 1 wrote: “L275 – “We finally check that considering different ways to characterise marine areas...” would maybe read better as “We finally check if different characterisations of marine areas...””

This was changed directly in the revised text in line 337.

Reviewer 1 wrote: “L277 – similarly to above instead of “8 ways” maybe it would be better to say “8 characterisations””

This was modified in the revised text in line 339.

Reviewer 1 wrote: “L296 – It might be good to explain the mid-domain effect at least briefly”

Mid-Domain Effect is explained in *Category 1 of Supplementary Text 1*. Nevertheless, we added “(formation of a peak in biodiversity at the middle of a latitudinal range)” in line 360.

Reviewer 1 wrote: “L296-299 – This paragraph reads as an introduction to the following paragraph, therefore should be incorporated accordingly”

We have merged this paragraph and the subsequent one in the revised text in lines 359 to 376.

Reviewer 1 wrote: “L313-L317 – This is a very long sentence. I am not sure what “the latter” in L317 refers to.”

We modified the text lines 390 to 396. We now write: “We are aware that origination/extinction, historical/time hypotheses and species interaction are important in terms of generation of new species and species saturation⁹. Despite the non-inclusion of these mechanisms in our model, however, our simulations capture both large-scale biodiversity patterns and global marine biodiversity changes. This result suggests that those mechanisms are not leading global biodiversity and its spatial arrangement during the Phanerozoic.”.

Reviewer 1 wrote: “L320 – “present” is used twice”

We deleted the second “at present” occurrence in this sentence, line 398.

Reviewer 1 wrote: “L343 – I am not quite sure why other studies focusing on biodiversity changes in specific groups are mentioned in this section. Are the findings of these studies significantly different from the patterns uncovered in this study?”

To clarify this point, we have added the following sentence in the revised text in lines 424 to 426: “Future versions of the macroecological model might provide specific biodiversity patterns for each of these groups.”.

Reviewer 1 wrote: “Figure 1.e – the text does not look aligned, maybe it has been aligned by the “centre” but it might look better if it’s just aligned to the left”

It is effectively aligned by the centre. We changed “0.9” to “0.90” in the figure to fix this problem.

Reviewer 1 wrote: “Figure 3. - As the second y-axis shows the same type of value in every subplot just a single “Fossil species richness index” label is needed”

This was modified in the revised version of the manuscript.

Reviewer 1 wrote: “Figure 4. legend – in c) it is not totally clear what density distribution refers to (geographical cells?)”

We added “(geographical cells)” in the figure caption.

Reviewer 1 wrote: “Methods - L88-L99 – it is not completely clear how the updated version of SNCI differs from the previous one”

We added “While all possible ecological niches are generated in SNCI-v1, the second version (SNCI-v2) directly considers a finite pool of randomly-generated niches, reducing the processing time.” in lines 101 to 103.

Reviewer 2

Reviewer 2 wrote: “Balembois et al. Undertake a project to look at what controls the Latitudinal Biodiversity Gradient (LBG) over the Phanerozoic and long-term changes, using a simulated approach that is then ground-truthed using fossil data. Research about the LBG over the Phanerozoic has gained a lot of interest over the last few years with increasing studies, which have mostly focused on fossil data rather than also simulating hypothetical changes. Overall, I must say I thought the article was excellently written and very easy to follow, and the last section “Limitations of our framework” also captures my main comments. I guess the question is are the limitations too big or are they reasonable. I would view them as future research directions that cannot be fixed quickly, and therefore should not stop this article from being published. I have no major criticisms of the manuscript. I have included below some thoughts I noted as I went through the article, but they are not suggestions for making corrections, more here are some comments and decide if you need to make changes according to them. I should also add that it appears that this model is founded under the idea that a model based on allopatric speciation is acceptable for testing the hypothesis, and I am not qualified to judge this decision.”

We thank Reviewer 2 for his support.

Reviewer 2 wrote: “Abstract: It is really strange to see so many references in an abstract. I think they should all be removed and then the ideas should be repeated in the introduction.”

We agree. We decided to delete all the references in the abstract of the revised manuscript.

Reviewer 2 wrote: “Line 26: “whether a single mechanism explains temporal trends and...”. Would you ever expect or believe a study that suggests one mechanism can explain controls on biodiversity? Even in this study it uses an approach that is a model of multiple (not a single) mechanism.”

We changed “single mechanism” to “primary cause” in line 27 and “primary” to “fundamental” in line 36. Indeed, in 1992 Klaus Rohde in his article “Latitudinal gradients in species diversity: the search for the primary cause.” suggested that global biodiversity patterns may be governed by a primary cause, and while some authors agree on a combination of multiple factors, various papers still try to identify a primary cause to global species arrangement.

Reviewer 2 wrote: "Line 70-71: You look at changes in diversity over the Phanerozoic. Have you ever considered starting in the Ediacaran, especially with the new view that the Cambrian explosion is a series of events that started before the Phanerozoic..."

Reviewer 2 probably refers here to the Avalon explosion and the supposed emergence of complex forms of life. Unfortunately, global palaeogeographical reconstructions and associated general circulation model simulations were not available to us for the Ediacaran, and fossil data are too laconic to allow robust estimates of biodiversity, hence reliable comparison between our simulations and observations for that period.

Reviewer 2 wrote: "Line 104: Foraminifera are not invertebrates, they are protists. Also seemed odd to not make a comparison to molluscs, bryozoans, echinoderms and brachiopods that have such a rich fossil record."

We changed "invertebrates" for "protists" in line 123. In Fig. 1, we only considered data from Tittensor and colleagues (2010) to compare present-day global spatial biodiversity patterns between observations and simulations and to check our model's performance in present day. Data of Tittensor and co-workers do not include the mentioned taxa.

Reviewer 2 wrote: "Line 114: You could have used Triton for investigating the Cenozoic trends in microfossil and the LBG beyond the continental margins."

We read the paper of Fenton *et al.*, 2021. However, as Triton database only covers the Cenozoic *i.e.* the last 66 million years, and as our study concerns the whole Phanerozoic, we needed fossil databases with a higher time coverage to allow comparisons with our model simulations.

Reviewer 2 wrote: "Line 123: 109 "snapshots" for 541 million years is still quite coarse. I think a higher resolution should be possible."

To our knowledge, there is no other set of worldwide paleoclimatic reconstructions for the whole Phanerozoic with similar spatial resolution and better temporal resolution. In our study, we used palaeo-reconstructions published in Valdes *et al.*, 2021, which were extremely computationally expensive to generate and cannot be extended. Each of the 109 simulations is run for about 12,000 years. As this is a fully complex climate model, this is probably unprecedented in its scale. Running on parallel cores on an HPC machine, each simulation runs at ~50 model years per real day, meaning that each simulation takes about 8 months to run. A few can be run simultaneously, but the overall effort to complete these simulations has taken about 5 years of real time.

Reviewer 2 wrote: "Line 126: "good job" is a subjective interpretation and funny to see in a scientific article. Could you include a statistical test?"

We replaced “*do a good job of reproducing*” by “*are significantly correlated positively with*” in line 155 and we mentioned “*Fig. 3a-b*” in lines 156 and 157 as Figure 3b includes the correlation value between observations and model reconstructions.

Reviewer 2 wrote: “Line 129-130. Only one of the databases in this list is actually up-to-date (PBDB) all of the others are out-of-date and have associated issues that are well-described in the literature. Why not just do the recent download of the PBDB.”

We wrote the following “*Fossil databases*” section in lines 152 to 173 of the revised methods: “*Amongst databases used in this paper, the oldest one is the Sepkoski’s database²⁵. It focuses on marine biodiversity, especially ichnospecies in marine facies, invertebrate species, marine genera and subgenera and marine metazoan families, through the last 500 Myrs and thus mixes several taxonomic levels. Alroy’s database²⁶ focuses on marine invertebrates genera across the Phanerozoic and is sometimes seen as an update of Sepkoski’s database, as it contains more fossils and more sampling conditions. It is also a precursor of the Paleobiodiversity database (PBDB). The updated version of Sepkoski’s database has been provided by the École Normale Supérieure de Lyon and may be found directly on their website (<https://acces.ens-lyon.fr/acces/thematiques/limites/paleobiodiversite/developper/banque-de-donnees-sepkoski>). It considers the same marine taxonomic levels as Sepkoski’s original database. The PBDB gathers palaeontological data from multiple sources and considers marine species only all across the world for all geological periods. It has been directly downloaded from the PBDB website (<https://paleobiodb.org/classic/displayDownloadGenerator>) and adapted to fit this study’s timeline. Zaffos’ database²⁷ also emanates from the PBDB but only considers marine animal genus richness, which makes comparison with our simulations more reliable. It was originally created to allow long-term comparison with continental fragmentation through the Phanerozoic. While the PBDB is the most recent and updated database here, it is also the most generalist one and it does not necessarily best fit our numerical experimentation conditions. Moreover, it also includes some biases, for example in preservation and sampling. This is why we chose to combine those databases and to recalculate a fossil species richness index through the Phanerozoic.” And we added the following sentence in lines 157 to 159 of the revised manuscript: “*This operation also allows us to combine recent and updated but generalist databases with others that better match our numerical experimentation conditions (Methods).*”.*

We also changed the previous version of the PBDB used in the manuscript and downloaded a new version focused on marine species only. We modified correlations in the revised manuscript accordingly. However, as stated above, we decided to keep all fossil databases to reduce biases inherent in each database.

Reviewer 2 wrote: “Line 140-141: This statement seems odd for the Cretaceous, because compared to older rocks, it is quite well-sampled.”

Several studies refer to possible biases in fossil preservation and sampling. In our manuscript we mentioned Flannery Sutherland and colleagues’ paper (2019), precisely entitled “*Does exceptional preservation distort our view of disparity in the fossil record?*”.

In the revised manuscript, we have also added the review from Nanglu and Cullen (2023) about “*sampling, preservational, analytical, and anthropogenic biases in fossil data across macroecological scales*” to strengthen this point.

Reviewer 2 wrote: “Line 147-150: Do you really believe that mass extinctions, which are partially defined as altering the trajectory in the evolution of life, do not play a first-order role in global biodiversity? I find this statement hard to digest. The modern oceans are more diverse because the Paleozoic fauna were eliminated at the Permian/Triassic boundary and some Mesozoic groups at the Cretaceous/Paleogene boundary. See also what you wrote in Line 205-207.”

We indeed found that, at the temporal resolution of our study, mass extinctions did not critically alter the trajectory of marine biodiversity in terms of global species richness and large-scale biodiversity patterns. Nevertheless, we know they have an effect on global assemblages and taxa composition through deep time, although these parameters are not investigated in the present work. We have clarified this point in the revised manuscript in lines 179 to 181: “*Although mass extinctions play a well-known role in the evolution of life and therefore in the composition of communities¹³, ...*”.

Reviewer 2 wrote: “Line 176-179: Given the experiment set up, this result is a function of the model, right?”

While we found a highly significant positive correlation ($r=0.90$, $p<0.01$, $p_{ACF}<0.01$) between modelled species richness with speciation and the available area around continents, we also found a significant positive correlation ($r=0.73$, $p<0.01$, $p_{ACF}=0.04$) between observed global species richness and available area, independently of our macroecological model. As our model without speciation does not fit the long-term area around continents neither fossil observations, we can deduce that allopatric speciation played a strong role in global marine biodiversity during the Phanerozoic.

Unravelling the drivers of marine biodiversity across the Phanerozoic

Alexis Balembois^{1*}, Alexandre Pohl², Bertrand Lefebvre³, Thomas Servais⁴, Daniel J. Lunt⁵, Paul J. Valdes⁵, Grégory Beaugrand^{1*}

¹ Univ. Littoral Côte d'Opale, CNRS, Univ. Lille, UMR 8187 LOG, F-62930 Wimereux, France

² Biogéosciences UMR 6282, Université Bourgogne Europe, CNRS F-21000 Dijon, France

³ Université Claude Bernard Lyon 1, ENSL, CNRS, LGL-TPE, F-69622, Villeurbanne, France

⁴ Univ. Lille, CNRS, ULR 8198-Evo-Eco-Paleo, F-59000 Lille, France

⁵ School of Geographical Sciences, University of Bristol, Bristol BS8 1SS, UK

* Corresponding authors: alexis.balembois@univ-littoral.fr, gregory.beaugrand@cnrs.fr

Point-by-point response to Reviewers' comments:

Reviewer 1

Reviewer 1 wrote: “*The authors have addressed all my concerns appropriately and I have no further suggestions. Congratulations!*”

We thank Reviewer 1 for her/his support and for comments that definitively helped to improve the quality of the manuscript.

Reviewer 2

Reviewer 2 wrote: “*I have gone through the revised submission and only have a couple tiny correction to highlight, I will also take the opportunity to respond to two rebuttal comments I was not satisfied with, although I do not request further edits as it won't change the article:*”

We thank Reviewer 2 for her/his thorough examination of our manuscript and her/his precious feedback that gave us the opportunity to clarify our paper.

Reviewer 2 wrote: “line 26: is "drive" the right verb? It sounds funny.”

We changed “*drive*” to “*combine*” in line 22.

Reviewer 2 wrote: “line 31: you are missing a comma before the word hence, otherwise you cannot read the sentence.”

We added a comma before the word “*hence*” in line 27.

Reviewer 2 wrote: “line 132: you need a space in-between climaticsimulations.”

We added a space in-between “*climatic simulations*” in line 129.

Reviewer 2 wrote: “*In the rebuttal you said that you will still use all the fossil databases because they have different biases and by including them you can address this somehow. I find that untrue, first you have the Sepkoski database (and its update), then the Alroy one, and then the PBDB, these are updates on one another fixing issues in the previous whilst providing more data. They all share exactly the same biases, but including them all you are essentially looking at how different amounts of data affect your results, but you could do this by subsampling the PBDB. What Jack Sepkoski and John Alroy achieved is truly phenomenal, but the PBDB is the database to use. P.S., the PBDB is a mix of everything ichnospecies and pollen included.*”

We have taken note of the referee’s point and would be happy to discuss it further in a few contribution to determine the best way of subsampling the PBDB. Another way to correct the dataset would be to combine theoretical approaches or ecological niche models to reanalyse the PBDB database. As part of a recent workshop, we discussed this issue, which might lead to a further research programme.

Reviewer 2 wrote: “*The second comment I was surprised by was saying that the computer simulations require a lot of time and therefore you cannot provide a higher resolution. The authors come from top universities with great computing facilities and expertise. There is of course a way, through more efficient programming or technological upgrades, that more simulations could be run. I also believe that you are genuinely making the simulations more efficient to run (behind the scenes) as any goal of a computer programmer.*”

We agree with the referee. However, with our current computers, we are afraid to say that this is the computational time needed currently. We might soon have more efficient computers.

Reviewer 2 wrote: “*Anyway, it is a good article that deserves to be published.*”

We thank again Reviewer 2 for her/his support.

Reviewer 2 wrote: “*The authors have done a good job of including the code and providing a good README file. I did not install and test the code.*”

Thank you.